# Sociodemographic and mental health predictors of mental health service use across provider types

Nelson Pang[1]*, Jessie Yeung[2]

1 Factor-Inwentash Faculty of Social Work, University of Toronto, Toronto, Ontario, Canada, 2 Department of Statistical Sciences, University of Toronto, Toronto, Ontario, Canada

* nelson.pang@mail.utoronto.ca

## Abstract

### Objective

To examine trends in mental health service use across four provider types (family doctors, psychiatrists, psychologists, and social workers) and identify sociodemographic predictors of provider-specific access in Canada.

### Methods

This study analyzes seven cycles (2007–2020) of the Canadian Community Health Survey, a nationally representative cross-sectional survey. Trends over time were examined using weighted proportions and counts of service users. Weighted multivariable logistic regression models were applied to the 2019–2020 cycle to assess associations between sociodemographic factors and provider-specific service use.

### Results

Family doctors were consistently the most accessed providers for mental health concerns, followed by psychologists and social workers, with psychiatrists being least accessed. Psychologist and social worker use increased between 2017 and 2019. In the adjusted regression models (2019–2020), women had higher odds of using family doctors (AOR = 1.21, 95% CI: 1.05–1.39) and social workers (1.19, 1.02–1.40) and lower odds of psychiatrists (AOR = 0.66, 95% CI:0.55–0.79) than men. Adults 65 + had greater odds of family-doctor use (AOR = 4.82, 95% CI: 3.59–6.47) and lower odds of psychologist (AOR = 0.33, 95% CI: 0.24–0.45) and social-worker use (AOR = 0.21, 95% CI:0.16–0.29) than ages 12–17. Post-secondary education (vs less than secondary school) was associated with higher psychologist use (AOR = 1.83, 95% CI 1.12–2.98). Higher income (≥$80,000 vs <$20,000) was associated with lower psychiatrist use (AOR = 0.66, 95% CI 0.50–0.86). Non-Indigenous respondents more

**Data availability statement:** The data used in this study are third-party data owned by Statistics Canada and cannot be shared by the authors. We analyzed the Canadian Community Health Survey (CCHS), Annual Component – Public Use Microdata Files (PUMF) for cycles 2007–2008, 2009–2010, 2011–2012, 2013–2014, 2015–2016, 2017–2018, and 2019–2020. Files were obtained via Odesi (Open Data Documentation, Extraction Service and Infrastructure) through the University of Toronto Libraries under the Data Liberation Initiative (DLI) license. Researchers at DLI-participating institutions can access the same PUMFs through their library's Odesi/Dataverse portal (https://odesi.ca). Others may obtain CCHS PUMFs from Statistics Canada's data portal (https://www150.statcan.gc.ca) or apply for access to the confidential master files via the Canadian Research Data Centre Network (CRDCN) (https://crdcn.ca), subject to approval and confidentiality requirements. The authors had no special access privileges that others would not have.

**Funding:** The author(s) received no specific funding for this work.

**Competing interests:** The authors have declared that no competing interests exist.

often used psychologists (AOR = 1.58, 95% CI: 1.13–2.23), and respondents who are not a visible minority more often used family doctors (AOR = 1.37, 95% CI:1.06–1.77).

## Conclusion

This study reveals a stratified mental health care system in Canada, where sociodemographic factors shape who accesses which providers. While primary care dominates, growth in psychologist and social worker use suggests shifting patterns of engagement. Findings underscore the need for policies that address financial and structural barriers, promote equitable access, and expand coverage for community-based mental health providers.

## Background

Mental health is a significant public health concern in Canada, with approximately 1 in 5 Canadians experiencing a mental illness every year [1]. Despite this access to mental health care is limited with studies estimating that only around 40–60% of people with a mental health problem seek professional assistance [2,3]. In 2018, an estimated 5.3 million Canadians reported needing help for their mental health in the previous year highlighting the widespread demand for mental health support across the Canadian population [4]. Alarmingly, nearly half of these individuals had their needs only partially met (1.2 million) or completely unmet (1.1 million), suggesting substantial gaps in service delivery for mental health. [4]

In Canada, mental health care is provided by a diverse range of professionals, including psychiatrists, psychologists, social workers, nurses, and physicians [3]. These different providers differ not only in their training and scope of practice but also in funding and accessibility [5]. For example, family physicians and psychiatrists are typically publicly funded and can be billed directly to provincial and territorial health insurance plans whereas services provided by psychologists and social workers may require out-of-pocket payment or private insurance coverage [3]. These differences in cost of services may result in financial barriers for individuals seeking care. Furthermore, these differences have implications on who receives care, what kind of care is delivered, and how barriers to access may differ across populations. In recent years, the mental health service landscape in Canada has become increasingly complex with expanding roles in the mental health care for non-physician providers such as psychologists and social workers [5]. This change reflects both efforts to meet growing demand for mental health care, shifts toward evidence-based care, and interprofessional models of service delivery [6,7].

Previous research has shown that mental health care utilization is shaped by an interplay of need, enabling factors and predisposing factors, such as symptom severity, income, gender, education, cultural background, wait time, and costs [8–12]. Despite Canada having universal health insurance, disparities and inequities are still associated with the underutilization of mental health care services [8,11,13,14]. For example, previous research in Canada has found that individuals with higher

education are more likely to utilize mental health services than those in less educated groups [11,13]. Likewise, previous research has found income-based inequity in access to mental health services [8,14]. Understanding which populations access specific types of providers is critical for informing mental health care and equity-focused policies. For example, investments in community-based providers such as psychologists and social workers may reduce reliance on primary care for mental health, improve continuity of care, and expand culturally appropriate service options [15–17].

A limitation in this research however is that most studies aggregate all types of mental health providers into one variable limiting the examination of differences in who accesses which services and under which circumstances. As a result, we lack nuanced understanding of how sociodemographic factors shape engagement with specific provider types, such as psychologists or social workers, whose roles and funding structures differ significantly from psychiatrists and family doctors. Furthermore, few studies have examined national level trends over time in provider specific mental health service use limiting our ability to assess how patterns have shifted in response to evolving service landscapes, policy changes, or public attitudes towards mental health. The data used in this study predate the COVID-19 pandemic, which led to rapid expansion of virtual mental health care and changes in service delivery across provider types. While these changes may have changed some patterns of mental health care access, pre-pandemic data remain important for establishing baseline trends and identifying structural inequities in mental health service use. Understanding these baseline patterns is essential for interpreting how access may have changed in the post-COVID-19 pandemic context.

To address these gaps, this study focuses on examining how sociodemographic and mental health characteristics are associated with mental health service use across four provider types (family doctors, psychiatrists, psychologists, and social workers) and trends in their use over time in Canada. By disaggregating service use by provider type, this research aims to uncover distinct access patterns across sociodemographic groups and inform efforts to promote equity in mental health care.

## Methods

### Study design and data source

We conducted a retrospective secondary analysis of seven cycles, using seven cycles (2007–2020) of the Canadian Community Health Survey (CCHS) public-use microdata. The CCHS is a nationally representative cross-sectional survey administered by Statistics Canada. The CCHS is a self-reported survey on health, healthcare utilization, and health determinants of the Canadian population aged 12 and above. The CCHS covers the population 12 years of age and over who live in Canada. Excluded from the CCHS are institutionalized residents, full-time members of the Canadian Armed Forces, residents of certain remote regions, or people living on reserves/other Indigenous settlements in the provinces. Altogether, these exclusions represent less than 3% of the target population. Sample sizes varied across cycles, and all analyses applied survey weights provided by Statistics Canada to ensure representativeness of the Canadian population. This study summarized trends of provider specific mental health access across 2007–2020 and provider specific logistic regressions were restricted to 2019–2020 because it was the most recent cycle and contained key predictors specified a priori for multivariable regression. See Appendix 1 for details of variable availability and wording between CCHS cycles.

### Ethics, data access, and participant confidentiality

This study used retrospective survey data from the CCHS Public Use Microdata Files (PUMFs), accessed for research purposes in January 2025. These data are collected and publicly released by Statistics Canada, which obtains informed consent from participants at the time of original data collection. For respondents aged 12–17 years, participation requires parental or guardian consent in addition to youth assent, in accordance with Statistics Canada data collection procedures. The PUMFs are fully anonymized prior to release and do not contain any identifying information. As a result, the authors

 

did not have access to any information that could identify individual participants during or after data collection, and additional institutional ethics approval was not required for this secondary analysis.

## Outcome variables

The data was subsetted to only include respondents who have consulted a mental health professional in the past 12-months (n = 13,580/weighted n = 4,679,132.44). The primary outcome were binary indicators of if respondents reported having seen or talked to a specific health professional based on the question *"In the past 12 months have you seen or talked to a health professional about your emotional or mental health?"*. Response options in the CCHS are provided as predefined categories for different provider types (i.e., family doctor, psychiatrist, psychologist, social worker, nurse, and other health professionals), and respondents may select multiple providers. This analysis focused on the four most common providers for mental health care (family doctor, psychiatrist, psychologist, and social work) to ensure interpretability and comparability across cycles. Each provider was coded as a separate variable as respondents could see more than one provider.

## Predictor variables

A variety of sociodemographic and mental health characteristics were included as predictors in the model. Sociodemographic characteristics included sex, age, household income, being a visible minority, education, immigration status, and Aboriginal identity. Being a visible minority is defined by Statistics Canada, in accordance with the Employment Equity Act, as persons, other than Indigenous peoples, who are non-Caucasian in race or non-white in colour, based on self-identification. Mental health characteristics included perceived health and perceived mental health.

## Statistical analysis

An exploratory descriptive analysis was conducted to examine trends in mental health service use across provider types over time (2007−2020). To identify factors associated with provider-specific mental health service we fit four separate, survey-weighted multivariable logistic regression models using the 2019−2020 cycle of the data for each provider type (family doctor, psychiatrist, psychologist, and social worker). The models access the relationship between the use of a specific provider adjusting for all other covariates. Each model included the same a priori covariates listed above to keep models comparable. Outcomes were binary (any past-12-month contact with the provider: yes/no). We modeled providers separately because respondents may see multiple providers and access differs for various reasons including pathways and financing. A single multinomial model would also impose inappropriate mutual exclusivity. Two-tailed tests with a threshold of α = 0.05 were used for all statistical tests. All analyses were conducted using R statistical software and survey weights were applied to account for the complex sampling design of the CCHS and to be representative of the Canadian population.

## Results

Between 2007 and 2020, family doctors consistently accounted for the highest proportion of mental health service use in Canada in every cycle with over half of service users responding to having accessed a family doctor for mental health care (See Fig 1). Across all cycles, the rank order of providers was stable and the patterns of service use appeared relatively stable over time based on descriptive trends, although no formal statistical tests of differences across years were conducted.. The proportion of people who accessed care through psychologists and social workers showed a gradual increase over time, while the proportion of visits to psychiatrists remained relatively unchanged over time. A one-year divergence appears in 2017, with a temporary spike for psychologist and dips slightly for the other providers but by the next cycle (2019) the pattern returns with psychologist and social workers converging at similar levels. We interpret 2017

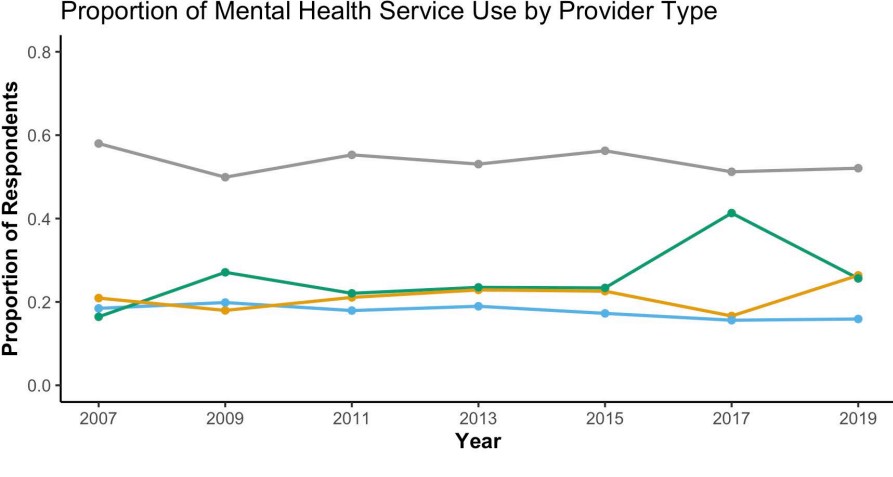

**Fig 1. Past-12-month contact for mental health by provider, CCHS 2007–2019 (survey-weighted proportion).**

as single cycle fluctuation rather than a break in trend. Estimates are survey-weighted and reflect that provider contacts are not mutually exclusive.

When examining weighted estimates of the number of Canadians accessing care, similar trends were observed (See Fig 2). The total number of individuals seeing family doctors for mental health reasons far exceeded those accessing psychologists, psychiatrists, or social workers. However, all provider types showed increases in absolute numbers over time reflecting increased mental service use overtime. Notably, a temporary drop in service use was observed in 2017 with a rebound by 2019 across all provider types, potentially reflecting data collection limitations in that survey cycle. Given each

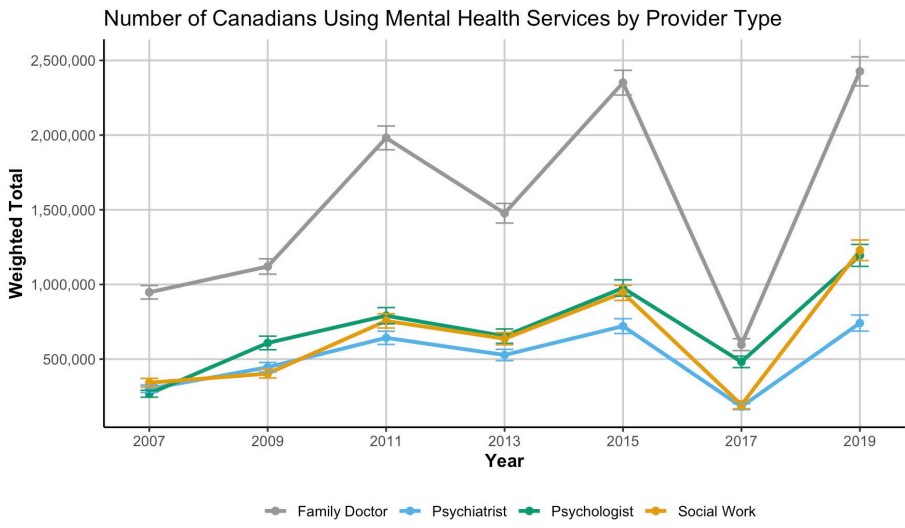

**Fig 2. Weighted number of Canadians reporting past-12-month mental-health contact by provider, CCHS 2007–2019.**

cycle of the CCHS has different weight calibrations and sample allocations, we believe this is a cycle level fluctuation rather than a structural change to mental health care use.

Table 1 presents the weighted characteristics of respondents who reported consulting a mental health professional in the past 12 months. The sample was predominantly women (65.9%) and individuals aged 18–49 years. Most respondents reported higher household incomes (≥$80,000; 60.1%) and post-secondary education (87.1%). The majority identified as non-visible minorities (86.2%) and non-immigrants (83.7%), with 5.0% identifying as Indigenous. Most participants reported good or very good general (67.5%) and mental health (62.2%). Adjusted associations between sociodemographic and mental health characteristics and provider-specific mental health service use are presented in Table 2. Because outcomes were modeled separately the estimates below should be interpreted as provider-specific associations.

Use of social workers was higher among women (AOR = 1.19, 95% CI: 1.02–1.40) and decreased with age across all groups relative to the 12–17 group (AOR [18–34] = 0.49, 95% CI: 0.38–0.63; AOR[35–49] = 0.38, 95% CI: 0.30–0.49; AOR [50–64] = 0.25, 95% CI: 0.18–0.33; AOR [65+] = 0.21, 95% CI: 0.16–0.29). Better perceived mental health was associated with lower odds compared to having poor perceived mental health (AOR [good] = 0.65, 95% CI: 0.49–0.87; AOR [very good] = 0.63, 95% CI: 0.46–0.86; AOR [excellent] = 0.64, 95% CI: 0.44–0.94).

Use of family doctors was higher among women than men (AOR = 1.21, 95% CI: 1.05–1.39) and increased sharply with age (vs. 12–17: AOR [18–34] = 2.97, 95% CI: 2.26–3.91; AOR[35–49] = 3.74, 95% CI: 2.86–4.90; AOR [50–64] = 3.98, 95% CI: 3.01–5.26; AOR [65+] = 4.82, 95% CI: 3.59–6.47). Higher household income was associated with higher odds of using a family doctor (vs. No income or less than $20,000. AOR[$40–59k]: 1.53, 95% CI:1.15–2.03; AOR[≥$80k]= 1.41, 95% CI: 1.11–1.80), whereas post-secondary education was associated with lower odds compared to those with less than secondary school graduation (AOR = 0.61, 95% CI: 0.45–0.83). Not being a visible minority was also associated with higher odds of seeing a family doctor (AOR: 1.37, 95% CI: 1.06–1.77). Better perceived mental health related to lower use compared to having poor perceived mental health (AOR [very good] = 0.57, 95% CI: 0.42–0.78; AOR [excellent] = 0.58, 95% CI: 0.41–0.84).

Odds of seeing a psychiatrist were lower for women than men (AOR = 0.66, 95% CI: 0.55–0.79) and lower for adults aged 35–49 relative to 12–17 (AOR = 0.62, 95% CI: 0.42–0.91). Higher income was associated with lower odds (vs. No income or less than $20,000. AOR [$40–59k] = 0.70, 95% CI: 0.50–0.97; AOR [$60–79k] = 0.60, 95% CI:0.42–0.85; APR [≥$80k] = 0.66, 95% CI: 0.50–0.86). Better perceived health and perceived mental health related to lower use (e.g., perceived health excellent vs poor perceived health. AOR = 0.34, 95% CI; 0.22–0.52; perceived mental health excellent. AOR = 0.24, 95% CI: 0.16–0.37).

Psychologist use declined at older ages (AOR [50–64] = 0.69, 95% CI:0.52–0.92; AOR [65+] = 0.33, 95% CI: 0.24–0.45) compared with the 12–17 age group. In contrast, post-secondary education was associated with higher odds (AOR = 1.83, 95% CI: 1.12–2.98) compared to those with less than secondary school graduation. Being non-indigenous was associated with higher psychologist used compared to indigenous groups (AOR: 1.58, 95% CI: 1.13–2.23). Excellent perceived health was also associated with higher odds compared to those with poor perceived health (AOR: 1.47, 95% CI: 1.01–2.15).

Across providers, immigrant status was not significantly associated with mental health service use. Household income showed no association with psychologist or social worker use. Being a visible minority and Indigenous identity showed no associations overall, with two exceptions (respondents who were not a visible minority having higher odds of family-doctor use and non-Indigenous respondents having higher odds of psychologist use).

## Discussion

This study examined national trends in mental health service use across four provider types (family doctors, psychiatrists, psychologists, and social workers) and identified sociodemographic predictors to accessing each provider using a recent nationally representative survey. Results show that family doctors were the most common provider for mental health,

**Table 1. Weighted characteristics of study sample (CCHS 2019-2020).**

| Characteristic | n (unweighted), % (weighted) |
|---|---|
| **Sex** | |
| Men | 4,303 (34.1%) |
| Women | 9,277 (65.9%) |
| **Age group** | |
| 12–17 | 914 (7.0%) |
| 18–34 | 3,257 (35.6%) |
| 35–49 | 3,463 (28.4%) |
| 50–64 | 3,219 (20.7%) |
| 65+ | 2,727 (8.3%) |
| **Household income** | |
| <$20,000 | 1,171 (5.9%) |
| $20,000–$39,999 | 1,919 (10.1%) |
| $40,000–$59,999 | 1,968 (12.5%) |
| $60,000–$79,999 | 1,570 (11.5%) |
| ≥$80,000 | 6,588 (60.1%) |
| **Visible minority status** | |
| Visible minority | 984 (13.8%) |
| Not a visible minority | 12,299 (86.2%) |
| **Highest level of education in household** | |
| Less than secondary school graduation | 612 (2.4%) |
| Secondary school graduation, no post-secondary | 1,673 (10.5%) |
| Post-secondary certificate/diploma or university degree | 10,475 (87.1%) |
| **Indigenous identity** | |
| Yes | 802 (5.0%) |
| No | 12,778 (95.0%) |
| **Immigrant status** | |
| Immigrant | 1,576 (16.3%) |
| Non-immigrant | 11,767 (83.7%) |
| **Perceived general health** | |
| Poor | 963 (5.3%) |
| Fair | 2,129 (13.0%) |
| Good | 4,569 (33.9%) |
| Very good | 4,225 (33.6%) |
| Excellent | 1,673 (14.1%) |
| **Perceived mental health** | |
| Poor | 1,047 (7.6%) |
| Fair | 2,899 (22.1%) |
| Good | 4,996 (37.0%) |
| Very good | 3,408 (25.2%) |
| Excellent | 1,192 (8.1%) |

Percentages are survey-weighted using CCHS weights. Unweighted total sample size = 13,580. Missing values were excluded from individual variables; therefore, category totals may not sum to the full sample size.

**Table 2. Adjusted associations between sociodemographic variables and mental health provider specific mental health service use.**

| | Social Work AOR (95%CI) | Family Doctor AOR (95%CI) | Psychiatry AOR (95%CI) | Psychologist AOR (95%CI) |
|---|---|---|---|---|
| **Sex** (Ref: Men) | | | | |
| Women | **1.19 (1.02–1.40)** | **1.21 (1.05–1.39)** | **0.66 (0.55–0.79)** | 0.98 (0.84–1.14) |
| **Age** (Ref: 12–17) | | | | |
| 18–34 | **0.49 (0.38–0.63)** | **2.97 (2.26–3.91)** | 0.82 (0.55–1.22) | 0.78 (0.59–1.03) |
| 35–49 | **0.38 (0.30–0.49)** | **3.74 (2.86–4.90)** | 0.62 (0.42–0.91) | 0.93 (0.71–1.22) |
| 50–64 | **0.25 (0.18–0.33)** | **3.98 (3.01–5.26)** | 0.83 (0.56–1.22) | **0.69 (0.52–0.92)** |
| 65+ | **0.21 (0.16–0.29)** | **4.82 (3.59–6.47)** | 0.96 (0.64–1.44) | **0.33 (0.24–0.45)** |
| **Household income** (Ref: <$20,000) | | | | |
| $20,000–$39,999 | 0.95 (0.69–1.31) | **1.37 (1.04–1.81)** | 0.82 (0.60–1.12) | 0.87 (0.62–1.23) |
| $40,000–$59,999 | 0.89 (0.65–1.22) | **1.53 (1.15–2.03)** | **0.70 (0.50–0.97)** | 0.93 (0.67–1.31) |
| $60,000–$79,999 | 0.82 (0.59–1.14) | 1.32 (0.98–1.77) | **0.60 (0.42–0.85)** | 1.14 (0.77–1.68) |
| ≥$80,000 | 0.90 (0.68–1.19) | **1.41 (1.11–1.80)** | **0.66 (0.50–0.86)** | 1.09 (0.82–1.46) |
| **Visible minority** (Ref: Visible minority) | | | | |
| Not a visible minority | 1.05 (0.79–1.41) | **1.37 (1.06–1.77)** | 1.23 (0.85–1.80) | 1.20 (0.88–1.63) |
| **Education** (Ref: Less than secondary school) | | | | |
| Secondary school | 0.96 (0.62–1.49) | 0.76 (0.54–1.08) | 1.12 (0.73–1.72) | 1.28 (0.74–2.24) |
| Post-secondary | 1.08 (0.72–1.62) | **0.61 (0.45–0.83)** | 1.40 (0.95–2.08) | **1.83 (1.12–2.98)** |
| **Indigenous identity** (Ref: Yes) | | | | |
| No | 0.84 (0.63–1.13) | 1.12 (0.84–1.49) | 1.10 (0.75–1.61) | **1.58 (1.13–2.23)** |
| **Immigrant status** (Ref: Immigrant) | | | | |
| Non-immigrant | 1.00 (0.77–1.29) | 1.00 (0.80–1.26) | 1.27 (0.95–1.70) | 0.93 (0.71–1.21) |
| **Perceived general health** (Ref: Poor) | | | | |
| Fair | 0.74 (0.53–1.04) | 0.90 (0.66–1.23) | **0.68 (0.48–0.94)** | 1.22 (0.85–1.76) |
| Good | 0.73 (0.53–1.00) | 1.02 (0.76–1.37) | **0.56 (0.41–0.77)** | 1.24 (0.89–1.72) |
| Very good | 0.73 (0.52–1.02) | 0.91 (0.67–1.24) | **0.39 (0.27–0.55)** | 1.33 (0.94–1.88) |
| Excellent | 0.72 (0.49–1.07) | 0.78 (0.56–1.11) | **0.34 (0.22–0.52)** | 1.47 (1.01–2.15) |
| **Perceived mental health** (Ref: Poor) | | | | |
| Fair | 0.79 (0.59–1.05) | 0.91 (0.68–1.23) | **0.52 (0.38–0.70)** | 0.93 (0.66–1.30) |
| Good | **0.65 (0.49–0.87)** | **0.74 (0.55–1.00)** | **0.33 (0.24–0.44)** | 0.92 (0.66–1.28) |
| Very good | **0.63 (0.46–0.86)** | **0.57 (0.42–0.78)** | **0.28 (0.20–0.41)** | 0.90 (0.63–1.28) |
| Excellent | **0.64 (0.44–0.94)** | **0.58 (0.41–0.84)** | **0.24 (0.16–0.37)** | 0.84 (0.56–1.26) |

AOR: Adjusted Odds Ratio; CI: Confidence Interval; **Boldface** indicates statistical significance at the 0.05 level.

followed by psychologists and social workers, while psychiatrists are accessed less frequently. These trends were generally stable over time, with increases in psychologist and social worker use observed between 2017 and 2019. Consistent with previous research, family doctors were found to be the most commonly accessed mental health care provider, reaffirming the central role of primary care in Canada's mental health system [5,18]. Given the high rates and numbers of people who seek mental health care from family doctors it is important that family doctors are trained in assessment and treatment of mental health [19]. Access to psychologists and social workers increased in recent years, particularly between 2017 and 2019, which may reflect changing public attitudes toward mental health, reduced stigma, and increased availability of services [20–22].

An important consideration in interpreting these findings is that the data reflect pre-pandemic patterns of mental health service use. The COVID-19 pandemic led to a multitude of changes in mental health care including the expansion of virtual care and telehealth services, which may have altered access to different provider types [23]. For example, virtual care may have reduced some geographic barriers while potentially introducing new challenges related to technology access, digital literacy, and privacy [23]. Despite these changes, many structural features of the Canadian mental health system such as differences in public versus private funding and provider availability remain largely unchanged. Because of this the patterns observed in this study likely reflect underlying systemic dynamics that continue to shape access to care. Likewise, these findings provide an important baseline for understanding how mental health service use has evolved in the post-COVID-19 pandemic era.

Logistic regression analyses from the 2019–2020 CCHS cycle indicate significant differences in access to providers across different demographic groups. Women had higher odds of seeking support from family doctors and social workers compared to men, aligning with prior literature showing greater mental health service use among women [24]. Age emerged as a significant predictor of provider type with older adults more likely to seek care from family doctors and younger individuals more likely to access psychologists and social workers. This trend may reflect generational differences in mental health literacy, familiarity with provider types, and access pathways [25,26]. Additionally, mental health services delivered by psychologists and social workers are not universally covered by provincial health plans in Canada, and younger individuals may access these providers through post-secondary institutions, school-based programs, or employee insurance plans [8,27]. The rise in mental health awareness among younger generations, combined with greater comfort in discussing psychological issues, may also increase their likelihood of seeking specialized care [28,29]. In contrast, older adults may be more affected by stigma surrounding mental illness and thus prefer family doctors who can address mental and physical health simultaneously in a less stigmatizing setting [26]. Furthermore, these age patterns likely also reflect differences in care continuity and navigation. Older adults have stronger ties to a usual family doctor and more multimorbidity which makes primary care a natural entry point for mental and physical concerns [26]. Younger adults in contrast more often have access to private insurance and Employee Assistance Programs (EAP) for counselling which may lower the threshold to see psychologist and social workers.

Income and education were associated with distinct patterns of provider use. Individuals with higher education were more likely to access psychologists, potentially reflecting the private-pay nature of these services and barriers related to insurance coverage [8,30]. In contrast, individuals with lower income and education levels were more likely to see family doctors or social workers, who are often covered under public or community-based services. These patterns highlight ongoing structural inequities in access to mental health care, where the ability to pay often determines the type and intensity of support received [8,30–32]. These findings support the need for publicly funded access to a broader range of providers to ensure that individuals are matched to appropriate services regardless of ability to pay. Specifically, in Canada psychologist and social workers are not universally covered and access often depends on employers benefits, EAPs, or private insurance which commonly cap the number of sessions [33]. By contrast, visits to family doctors and psychiatrist are publicly insured which may explain a shift towards these providers among lower income groups.

Self-perceived mental and general health status were also associated to provider use. Individuals who rated their mental health as poor had significantly higher odds of accessing all four types of mental health providers. However, the findings also reveal potential disparities in access as those who were not visible minorities had higher odds of accessing family doctors and non-Indigenous individuals were more likely to access psychologists. These results align with existing literature documenting racial and cultural inequities in mental health service access in Canada, driven by factors such as discrimination, mistrust of providers, culturally unsafe care, and underrepresentation of racialized and Indigenous practitioners [34–37]. Furthermore, these disparities are also shaped by geography and provider supply [38,39]. For example, mental health care providers are often concentrated in large urban centre resulting in longer travel times and indirect costs for rural communities to accessing care [40]. These findings underscore the importance of promoting equitable mental

health services and support community-based models of care that are grounded in trust and cultural responsiveness. Several actions could improve equitable access to mental health services in Canada. First, expanding publicly funded, evidence-based psychotherapy would reduce the role of financial barriers [38]. Second, integrating mental health care providers into primary care teams by streamlining referrals, provided shared care can result in shortened wait times and improved continuity in care [5,18]. Equity can be further strengthened through Indigenous-led services, cultural-safety standards, language interpretation, and navigation supports for newcomers and racialized communities. Furthermore, private insurance and EAPs need to be modernized to ensure adequate session coverage. Lastly, further investment needs to be made in community-based mental health care including walk-in counselling, youth hubs, peer and Indigenous led programs and system navigation would expand timely service and reduce avoidable reliance on emergency and specialist services [15,17]. These findings reveal an inequitable mental health care system in Canada, where sociodemographic factors such as age, gender, income, education, and racial identity shape not only access but also the type of provider individuals are able to see. While need-based access patterns were evident there are clear structural and systemic barriers that limit equitable engagement with care. These disparities reinforce the importance of expanding publicly funded mental health services beyond primary care and ensuring broader access to psychologists and social workers. As Canada continues to build toward a more integrated and accessible mental health system, efforts must prioritize reducing financial, geographic, and cultural barriers to ensure that all individuals can access the mental health care they need. Future research should explore the intersection of provider preferences, treatment experiences, and long-term outcomes to further inform changes to the mental health care system.

## Future directions

To build on these findings, future research should focus on using longitudinal or linked survey administrative data to map sequences of care, switching between providers, and duration/intensity of treatment. Similarly, methods that focus on concurrent mental health service provider use would be valuable allowing for the use of multinomial models and to better understanding how individuals are accessing the mental health care system. Equity should be a focus allowing exploration on intersectional effects across gender, age, income, education, racialization, immigration, Indigenous identity, and urbanicity. Likewise, further exploration is needed on needs and barriers including coverage, out-of-pocket costs, wait times, language, and providers availability. Lastly, qualitative research with equity seeking groups can illuminate preferences, experiences, and structural barriers that quantitative data alone may miss.

## Strengths

This study has several strengths. First, it leverages the Canadian Community Health Survey across seven cycles (2007–2020), yielding nationally representative estimates and trends, with Statistics Canada's sampling procedures and survey weights enhancing generalizability. Similarly, we use survey-weighted and fit provider-specific models (family doctor, psychiatrist, psychologist, social worker) allowing for a nuanced view of provider specific mental health care. Furthermore, the multivariable models are a priori and equity-focused, adjusting for correlated sociodemographic and health factors to explore variables related to policy.

## Limitations

This study has several limitations. First, mental health service use was based on self-report and may be subject to recall or reporting biases. Second, only service use within the past year was captured without accounting for frequency, duration, or perceived adequacy of care. Third, the regression models only used data from a single CCHS cycle (2019–2020). This is due to an inability to combine data from different cycles together as predictor variables were measured differently across various cycles of the survey. This means that logistic regression results are not able to describe longer-term

patterns. Additionally, service use was measured separately for each provider type, without information on concurrent or integrated care. Likewise, given we did not model concurrent use across providers the results do not capture substitution, sequencing, or intensity of multi-provider care. Furthermore, we did not formally test differences in service use across survey years a result interpretations of temporal trends are based on descriptive patterns. Additionally, the CCHS lacks questions on visit timing, referral pathways, and number of contacts, limiting our ability to characterize multi-provider pathways. Furthermore, we did not have measures of symptom severity, diagnosis, functional impairment, prior treatment, local provider supply, wait times, out-of-pocket costs, or insurance and EAP coverage. We also lacked information on language, travel time, and referral practices. The lack of these factors does not provide a full picture of mental health service use and the omission of these factors could confound associations if they correlate with both with both sociodemographic characteristics and provider use. Because of this our findings should be interpreted as adjusted associations rather than causal effects. Future studies with linked longitudinal data and methods that explicitly accommodate overlap is needed.

## Conclusion

This study provides national evidence on who accesses which mental health providers in Canada and how that varies over time. While primary care remains a central entry point for mental health services, the growing role of psychologists and social workers may reflect changes in public demand and service availability. However, significant disparities persist particularly in access to privately funded services. These findings underscore the need for policy efforts to improve equity in mental health care delivery across provider types and population groups.

## Supporting information

**S1 File. Appendix 1. Variable definitions and availability CCHS.**
(PDF)

## Author contributions

**Conceptualization:** Nelson Pang, Jessie Yeung.

**Formal analysis:** Nelson Pang, Jessie Yeung.

**Investigation:** Nelson Pang.

**Methodology:** Nelson Pang, Jessie Yeung.

**Visualization:** Jessie Yeung.

**Writing – original draft:** Nelson Pang, Jessie Yeung.

**Writing – review & editing:** Nelson Pang, Jessie Yeung.

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
