## [Decision Letter · Decision Letter 0]

11 Jul 2025

PONE-D-25-27752Sociodemographic and Mental Health Predictors of Mental Health Service Use Across Provider TypesPLOS ONE

Dear Dr. Pang,

Thank you for submitting your manuscript to PLOS ONE. After careful consideration, we feel that it has merit but does not fully meet PLOS ONE’s publication criteria as it currently stands. Therefore, we invite you to submit a revised version of the manuscript that addresses the points raised during the review process.

**Please provide a point-by-point response to all the reviewers' comments.**

We look forward to receiving your revised manuscript.

Kind regards,

Mohammad Mofatteh, PhD, MPH, MSc, PGCert, BSc (Hons), MB BCh (c)

Academic Editor

PLOS ONE

2. For studies involving third-party data, we encourage authors to share any data specific to their analyses that they can legally distribute. PLOS recognizes, however, that authors may be using third-party data they do not have the rights to share. When third-party data cannot be publicly shared, authors must provide all information necessary for interested researchers to apply to gain access to the data. (https://journals.plos.org/plosone/s/data-availability#loc-acceptable-data-access-restrictions)

Additional Editor Comments:

Please provide a point-by-point response to all the reviewers' comments.

Reviewers' comments:

Reviewer's Responses to Questions

**Comments to the Author**

1. Is the manuscript technically sound, and do the data support the conclusions?

Reviewer #1: Yes

Reviewer #2: Yes

2. Has the statistical analysis been performed appropriately and rigorously? 

Reviewer #1: No

Reviewer #2: Yes

3. Have the authors made all data underlying the findings in their manuscript fully available?

Reviewer #1: No

Reviewer #2: No

4. Is the manuscript presented in an intelligible fashion and written in standard English?

Reviewer #1: Yes

Reviewer #2: No

5. Review Comments to the Author

Reviewer #1: Hello dear authors.

MS Id: PONE-D-25-27752

Title: Sociodemographic and Mental Health Predictors of Mental Health Service Use Across Provider Types

Type: Research Article

Here are my recommendations about the mentioned MS:

Title:

• I suggest the title to be revised to “Mental Health Predictors of Mental Health Service Use Across Provider Types”.

Abstract:

• Remove the last part of methodology.

• Revise the results section in abstract for presenting the regression analysis.

Introduction:

• Adding reference for middle of the second paragraph.

• Provide a research gap and problem statement.

Methodology:

• Design of the study need to be mentioned the first.

• Inclusion and Exclusion Criteria not mentioned.

• Which type of logistic regression were used? Why? How?

• Univariate and multivariate logistic regression analysis not performed, why?

• Explain the logistic regression analysis in detail in the statistical analysis.

Results:

• Presenting the results in the table 1 not enough indicated.

• Revise the comments for logistic regression.

Discussion:

• Looks good.

Conclusion:

• Looks good.

References:

• Looks good.

Figures and tables:

• Figures looks good.

Some more issues should be considered necessary for publication:

• Suggestions for future studies also be mentioned.

• Please provide at least two related strengths for MS.

Reviewer #2: There are numerous grammatical errors and awkward sentence constructions. This research will addresses key knowledge gaps... should be corrected to This research addresses key knowledge gaps....Consider thorough proofreading or language editing to improve readability and professionalism.

The abstract is too long and includes excessive methodological detail. Condense the abstract and emphasize key findings and implications more succinctly.

You mention that logistic regressions were only done for 2019–2020 due to differences in variable measurement across survey cycles. However, you do not provide details or references showing which variables changed and how. Add an appendix table comparing variable definitions across cycles.

The exclusion of provider overlap in the logistic models is a serious limitation. Many people access multiple providers. Discuss this more explicitly, and consider sensitivity analyses or multinomial models if feasible.

Table 1 is dense and hard to interpret. Consider moving odds ratios and confidence intervals to supplemental material and highlighting only significant findings in the main text. You could benefit from a heatmap-style table or bolding to visually distinguish significant results.

The manuscript references Figures 1 and 2, but they are not embedded or summarized effectively in the results. Embed or summarize the main trends and include captions that explain anomalies (2017 drop).

The discussion covers many findings but often lacks depth in interpretation. Provide more nuance on why certain sociodemographic factors (e.g., education, income) are predictive of use for certain providers. Integrate more Canadian-specific policy recommendations and barriers (wait times, geographic access, EAP programs).

The limitations section is too brief. Elaborate on limitations of self-report data, cross-sectional design, and unobserved confounders (e.g., provider availability, wait times, severity of need).

While you mention Statistics Canada data, the Data Availability Statement is vague. Include exact dataset names, links to access procedures, and R code availability (if possible).

Include OR values in the abstract for the most important findings. Improve consistency in terminology — e.g., switch between visible minority and non-visible minority inconsistently.

6. PLOS authors have the option to publish the peer review history of their article (what does this mean?). If published, this will include your full peer review and any attached files.

Reviewer #1: **Yes:** Salar Omar Abdulqadir

Reviewer #2: No

---

## [Author Response · Author response to Decision Letter 1]

26 Aug 2025

Response to Reviewer for Manuscript: Sociodemographic and Mental Health Predictors of Mental Health Service Use Across Provider Types

Reviewer #1

Reviewer Comment: Title: I suggest the title to be revised to “Mental Health Predictors of Mental Health Service Use Across Provider Types”.

Author Response: Thank you for the suggestion. We have opted to use the original title, “Sociodemographic and Mental Health Predictors of Mental Health Service Use Across Provider Types” as it more accurately reflects the scope of the study. In addition to mental health characteristics, our analysis includes a range of sociodemographic variables (e.g., age, gender, income, education, Indigenous identity, and immigration status) as key predictors.

Reviewer Comment: Abstract: Remove the last part of methodology.

Author Response: Thank you for the suggestion. We have removed the last part of methodology from the abstract.

Reviewer Comment: Abstract: Revise the results section in abstract for presenting the regression analysis.

Author Response: The results in the abstract have been edited when presenting the regression analysis and now reads: In the adjusted regression models (2019–2020), women had higher odds of using family doctors (AOR = 1.21, 95% CI: 1.05–1.39) and social workers (1.19, 1.02–1.40) and lower odds of psychiatrists (AOR = 0.66, 95% CI:0.55–0.79) than men. Adults 65+ had greater odds of family-doctor use (AOR = 4.82, 95% CI: 3.59–6.47) and lower odds of psychologist (AOR = 0.33, 95% CI: 0.24–0.45) and social-worker use (AOR = 0.21, 95% CI:0.16–0.29) than ages 12–17. Post-secondary education (vs less than secondary school) was associated with higher psychologist use (AOR = 1.83, 95% CI 1.12–2.98). Higher income (≥$80,000 vs <$20,000) was associated with lower psychiatrist use (AOR = 0.66, 95% CI 0.50–0.86). Non-Indigenous respondents more often used psychologists (AOR = 1.58, 95% CI: 1.13–2.23), and respondents who are not a visible minority more often used family doctors (AOR = 1.37, 95% CI:1.06–1.77).

Reviewer Comment: Introduction: Adding reference for middle of the second paragraph.

Author Response: Thank you for the suggestion. We have added a reference for the middle of the second paragraph.

Reviewer Comment: Provide a research gap and problem statement.

Author Response: We have revised this section to provide a cleaer research gap and problem statement. This section now reads as: A limitation in this research however is that most studies aggregate all types of mental health providers into one variable limiting the examination of differences in who accesses which services and under which circumstances. As a result, we lack nuanced understanding of how sociodemographic factors shape engagement with specific provider types, such as psychologists or social workers, whose roles and funding structures differ significantly from psychiatrists and family doctors. Furthermore, few studies have examined national level trends over time in provider specific mental health service use limiting our ability to assess how patterns have shifted in response to evolving service landscapes, policy changes, or public attitudes towards mental health. To address these gaps, this study focuses on examining how sociodemographic and mental health characteristics are associated with mental health service use across four provider types (family doctors, psychiatrists, psychologists, and social workers) and trends in their use over time in Canada. By disaggregating service use by provider type, this research aims to uncover distinct access patterns across sociodemographic groups and inform efforts to promote equity in mental health care.

Reviewer Comment: Design of the study need to be mentioned the first.

Author Response: We now open the Methods with a clear statement of design and data source and now reads: We conducted a cross-sectional, population-based analysis using seven cycles (2007-2020) of the Canadian Community Health Survey (CCHS) public-use microdata. The CCHS is a nationally representative cross-sectional survey administered by Statistics Canada. The CCHS is a self-reported survey on health, healthcare utilization, and health determinants of the Canadian population aged 12 and above. The CCHS covers the population 12 years of age and over who live in Canada. Excluded from the CCHS are institutionalized residents, full-time members of the Canadian Armed Forces, residents of certain remote regions, or people living on reserves/other Indigenous settlements in the provinces. Altogether, these exclusions represent less than 3% of the target population. Sample sizes varied across cycles, and all analyses applied survey weights provided by Statistics Canada to ensure representativeness of the Canadian population. This study summarized trends of provider specific mental health access across 2007–2020 and provider specific logistic regressions were restricted to 2019–2020 because it was the most recent cycle and contained key predictors specified a priori for multivariable regression. See Appendix 1 for details of variable availability and wording between CCHS cycles.

Reviewer Comment: Inclusion and Exclusion Criteria not mentioned.

Author Response: Inclusion and exclusion criteria was added on the CCHS. This now reads: The CCHS covers the population 12 years of age and over who live in Canada. Excluded from the CCHS are institutionalized residents, full-time members of the Canadian Armed Forces, residents of certain remote regions, or people living on reserves/other Indigenous settlements in the provinces. Altogether, these exclusions represent less than 3% of the target population.

Reviewer Comment: Which type of logistic regression were used? Why? How?

Author Response: We clarified that we used survey-weighted multivariable logistic regression fitted separately for each provider and now reads: To identify factors associated with provider-specific mental health service we fit four separate, survey-weighted multivariable logistic regression models using the 2019-2020 cycle of the data for each provider type (family doctor, psychiatrist, psychologist, and social worker).

Reviewer Comment: Univariate and multivariate logistic regression analysis not performed, why?

Author Response: We did not perform univariate screening instead each provider-specific outcome was modeled with the prespecified multivariable set to avoid data-driven selection, preserve comparability across providers, and prevent exclusion of non-significant yet confounding covariates. Additionally, because key sociodemographic and health factors are correlated and rarely operate in isolation (e.g., income, education, racialization), multivariable models are required to estimate adjusted associations. This approach aligns with methodological guidance that discourages univariable screening in favor of theory-driven adjustment.

Reviewer Comment: Explain the logistic regression analysis in detail in the statistical analysis.

Author Response: We have added more detail into this statistical analysis and now reads as: An exploratory descriptive analysis was conducted to examine trends in mental health service use across provider types over time (2007-2020). To identify factors associated with provider-specific mental health service we fit four separate, survey-weighted multivariable logistic regression models using the 2019-2020 cycle of the data for each provider type (family doctor, psychiatrist, psychologist, and social worker). The models access the relationship between the use of a specific provider adjusting for all other covariates. Each model included the same a priori covariates listed above to keep models comparable. Outcomes were binary (any past-12-month contact with the provider: yes/no). We modeled providers separately because respondents may see multiple providers and access differs for various reasons including pathways and financing. A single multinomial model would also impose inappropriate mutual exclusivity. Two-tailed tests with a threshold of α = 0.05 were used for all statistical tests. All analyses were conducted using R statistical software and survey weights were applied to account for the complex sampling design of the CCHS and to be representative of the Canadian population.

Reviewer Comment: Presenting the results in the table 1 not enough indicated; Revise the comments for logistic regression.

Author Response: Thank you for this suggestion. We revised the Results to add clear, in-text interpretation of Table 1. We now summarize the key statistically significant findings in four short paragraphs organized by provider and include a brief non-findings paragraph to for simplicity. We also strengthened the Table 1 caption and notes.

Reviewer Comment: Discussion: Looks good.

Author Response: Thank you

Reviewer Comment: Conclusion: Looks good.

Author Response: Thank you

Reviewer Comment: References: Looks good.

Author Response: Thank you

Reviewer Comment: Figures and tables: Figures looks good.

Author Response: Thank you

Reviewer Comment: Some more issues should be considered necessary for publication: Suggestions for future studies also be mentioned.

Author Response: Thank you for this suggestion. We have a added a section on future directions for research that reads: To build on these findings, future research should focus on using longitudinal or linked survey administrative data to map sequences of care, switching between providers, and duration/intensity of treatment. Similarly, methods that focus on concurrent mental health service provider use would be valuable allowing for the use of multinomial models and to better understanding how individuals are accessing the mental health care system. Equity should be a focus allowing exploration on intersectional effects across gender, age, income, education, racialization, immigration, Indigenous identity, and urbanicity. Likewise, further exploration is needed on needs and barriers including coverage, out-of-pocket costs, wait times, language, and providers availability. Lastly, qualitative research with equity seeking groups can illuminate preferences, experiences, and structural barriers that quantitative data alone may miss.

Reviewer Comment: Please provide at least two related strengths for MS.

Author Response: Thank you for this suggestion. We have added a strength section that reads: This study has several strengths. First, it leverages the Canadian Community Health Survey across seven cycles (2007–2020), yielding nationally representative estimates and trends, with Statistics Canada’s sampling procedures and survey weights enhancing generalizability. Similarly, we use survey-weighted and fit provider-specific models (family doctor, psychiatrist, psychologist, social worker) allowing for a nuanced view of provider specific mental health care. Furthermore, the multivariable models are a priori and equity-focused, adjusting for correlated sociodemographic and health factors to explore variables related to policy.

Reviewer #2

Reviewer Comment: There are numerous grammatical errors and awkward sentence constructions. This research will addresses key knowledge gaps... should be corrected to This research addresses key knowledge gaps....Consider thorough proofreading or language editing to improve readability and professionalism.

Author Response: Thank you for reviewing this manuscript. We have gone through the manuscript for proofreading to improve readability.

Reviewer Comment: The abstract is too long and includes excessive methodological detail. Condense the abstract and emphasize key findings and implications more succinctly.

Author Response: We have edited to abstract to be more concise and focus on key findings.

Reviewer Comment: You mention that logistic regressions were only done for 2019–2020 due to differences in variable measurement across survey cycles. However, you do not provide details or references showing which variables changed and how. Add an appendix table comparing variable definitions across cycles.

Author Response: Thank you for this suggestion. We have added Appendix 1: Variable Definitions and Availability Across CCHS Cycles (2007–2020), which lists each construct and the availability of the variable in each cycle and any changes to the variable. Reasons for conducting logistic regressions on the 2019-2020 cycle include visible minority only being available in the 2019-2020 cycle, indigenous identity first appearing n the 2015-2016 cycle, and household income being measured differently prior to 2013. We also added a sentence in Methods directing readers to Appendix 1 for further details on the variables and the CCHS cycles.

Reviewer Comment: The exclusion of provider overlap in the logistic models is a serious limitation. Many people access multiple providers. Discuss this more explicitly, and consider sensitivity analyses or multinomial models if feasible.

Author Response: We agree that the concurrent use is important. We now state explicitly in the results and limitations that outcomes were modeled separately as any contact per provider. For example, in the results we have added: Because outcomes were modeled separately the estimates below should be interpreted as provider-specific associations. And in the limitation we have added: Likewise, given we did not model concurrent use across providers the results do not capture substitution, sequencing, or intensity of multi-provider care. Additionally, the CCHS lacks questions on visit timing, referral pathways, and number of contacts, limiting our ability to characterize multi-provider pathways. Furthermore, we did not have measures of symptom severity, diagnosis, functional impairment, prior treatment, local provider supply, wait times, out-of-pocket costs, or insurance and EAP coverage. We also lacked information on language, travel time, and referral practices. The lack of these factors does not provide a full picture of mental health service use and the omission of these factors could confound associations if they correlate with both with both sociodemographic characteristics and provider use. Because of this our findings should be interpreted as adjusted associations rather than causal effects. Future studies with linked longitudinal data and methods that explicitly accommodate overlap is needed.

We considered the suggested sensitivity analyses and multinomial models but did not implement them for methodological and practical reasons. For example, multinomial models required many overlap categories and with small cell counts this will lead to instability under the complex survey design.

Reviewer Comment: Table 1 is dense and hard to interpret. Consider moving odds ratios and confidence intervals to supplemental material and highlighting only significant findings in the main text. You could benefit from a heatmap-style table or bolding to visually distinguish significant results.

Author Response: Thank you for this helpful suggestion. We retained the full numeric Table 1 to preserve exact effect sizes and confidence intervals. To improve readability, we bolded all statistically significant AORs additionally we have updated the results section to focus on each model to ease with readability. These changes to Table 1 make it easier to read while maintaining transparency.

Reviewer Comment: The manuscript references Figures 1 and 2, but they are not embedded or summarized effectively in the results. Embed or summarize the main trends and include captions that explain anomalies (2017 drop).

Author Response: Thank you for this suggestion we have edited the text to better summarize the main trends. This section now reads for figure 1: Between 2007 and 2020, family doctors consistently accounted for the highest proportion of mental health service use in Canada in every cycle with over half of service users responding to having accessed a family doctor for mental health care (See Figure 1). Across all cycles, the rank order of providers was stable and the pattern over the years are relatively stable, with only modest year-to-year variation The proportion of people who accessed care through psychologists and social workers showed a gradual increase over t

---

## [Decision Letter · Decision Letter 1]

2 Apr 2026

PONE-D-25-27752R1

Sociodemographic and Mental Health Predictors of Mental Health Service Use Across Provider Types

PLOS One Dear Dr. Pang,

Thank you for submitting your manuscript to PLOS ONE. After careful consideration, we feel that it has merit but does not fully meet PLOS ONE’s publication criteria as it currently stands. Therefore, we invite you to submit a revised version of the manuscript that addresses the points raised during the review process.

The manuscript has been further evaluated by two reviewers, and their comments are available below.

Could you please carefully revise the manuscript to address all comments raised?

A letter that responds to each point raised by the academic editor and reviewer(s). You should upload this letter as a separate file labeled 'Response to Reviewers'.A marked-up copy of your manuscript that highlights changes made to the original version. You should upload this as a separate file labeled 'Revised Manuscript with Track Changes'.An unmarked version of your revised paper without tracked changes. You should upload this as a separate file labeled 'Manuscript'

We look forward to receiving your revised manuscript.

Kind regards,

Ilse Bloom

Staff Editor

PLOS One

Journal Requirements:

Reviewers' comments:

Reviewer's Responses to Questions

**Comments to the Author**

1. If the authors have adequately addressed your comments raised in a previous round of review and you feel that this manuscript is now acceptable for publication, you may indicate that here to bypass the “Comments to the Author” section, enter your conflict of interest statement in the “Confidential to Editor” section, and submit your "Accept" recommendation.

Reviewer #3: All comments have been addressed

Reviewer #4: (No Response)

2. Is the manuscript technically sound, and do the data support the conclusions?

Reviewer #3: Yes

Reviewer #4: Partly

3. Has the statistical analysis been performed appropriately and rigorously? 

Reviewer #3: Yes

Reviewer #4: Yes

4. Have the authors made all data underlying the findings in their manuscript fully available?

Reviewer #3: No

Reviewer #4: Yes

5. Is the manuscript presented in an intelligible fashion and written in standard English?

Reviewer #3: Yes

Reviewer #4: Yes

6. Review Comments to the Author

Reviewer #3: There are 2 occasions in which the authors stated that the study is cross sectional, whereas it is retrospective. This needs to be corrected

Reviewer #4: The current manuscript leverages large national Canada data over the years 2007-2020 to explore mental healthcare help-seeking by type of provider and illuminates potential disparities in regard to access to care. This is an important area of research with great implications for public health messaging and other interventions aimed at improving access. The current manuscript is a revision of a previous submission. I was not a reviewer of the previous draft, and so this is my first encounter with this work. Overall while the topic is important and the data sufficient to meet the research question, in my opinion there are several areas in need of revision.

A major issue that needs to be addressed in the Introduction and the Discussion is the impact of the COVID-19 pandemic on access to mental healthcare, particularly with regard to telehealth. The fact that all data precede what constituted a massive shift in service provision is a substantial limitation. The authors need to justify how and why these data are still relevant to the practice of mental healthcare in 2026.

Additional comments:

- The assent/consent process for youth aged 12-18 needs to be explained.

- Were response options for the outcome variable limited to the 4 provider types? Were there others? Or was it a free response that then was coded by researchers?

- How was “Visible minority” defined?

- I did not see any sample descriptives. This basic information is needed for readers to contextualize the findings.

- The authors interpret the pattern of service use over the years as “relatively stable.” is this a subjective observation or were differences between years tested? The latter would strengthen the manuscript.

- Table 1 is extremely hard to read and is not formatted following APA guidelines

- Several assertions are made in the Discussion with no accompanying reference.

7. PLOS authors have the option to publish the peer review history of their article (what does this mean?). If published, this will include your full peer review and any attached files.

Reviewer #3: No

Reviewer #4: No

---

## [Author Response · Author response to Decision Letter 2]

15 Apr 2026

Reviewer #3

Reviewer Comment: There are 2 occasions in which the authors stated that the study is cross sectional, whereas it is retrospective. This needs to be corrected

Author Response: Thank you for this comment. We agree that the manuscript should more clearly distinguish between the cross-sectional nature of the CCHS data and the retrospective nature of our analysis. We have revised the manuscript to describe the study as a retrospective secondary analysis of cross-sectional survey data where appropriate. For instance, the methods section now reads: “We conducted a retrospective secondary analysis of seven cycles, using seven cycles (2007-2020) of the Canadian Community Health Survey (CCHS) public-use microdata.”

Reviewer #4

Reviewer Comment: The current manuscript leverages large national Canada data over the years 2007-2020 to explore mental healthcare help-seeking by type of provider and illuminates potential disparities in regard to access to care. This is an important area of research with great implications for public health messaging and other interventions aimed at improving access. The current manuscript is a revision of a previous submission. I was not a reviewer of the previous draft, and so this is my first encounter with this work. Overall while the topic is important and the data sufficient to meet the research question, in my opinion there are several areas in need of revision.

Author Response: Thank you your review of this manuscript. Below, we address each of your comments in detail.

Reviewer Comment: A major issue that needs to be addressed in the Introduction and the Discussion is the impact of the COVID-19 pandemic on access to mental healthcare, particularly with regard to telehealth. The fact that all data precede what constituted a massive shift in service provision is a substantial limitation. The authors need to justify how and why these data are still relevant to the practice of mental healthcare in 2026.

Author Response: Thank you for this comment. We agree that the COVID-19 pandemic led to substantial changes in mental health service delivery, particularly through the expansion of telehealth, and that our data (2007–2020) largely reflect pre-pandemic patterns of care. While our study captures pre-pandemic service use, we argue that this data provides an important baseline for understanding structural patterns in access to different provider types, many of which (e.g., funding models and provider distribution) continue to shape access in the post-pandemic era.

We have added to the background section: “The data used in this study predate the COVID-19 pandemic, which led to rapid expansion of virtual mental health care and changes in service delivery across provider types. While these changes may have changed some patterns of mental health care access, pre-pandemic data remain important for establishing baseline trends and identifying structural inequities in mental health service use. Understanding these baseline patterns is essential for interpreting how access may have changed in the post-COVID-19 pandemic context.”

We have added to discussion section: “An important consideration in interpreting these findings is that the data reflect pre-pandemic patterns of mental health service use. The COVID-19 pandemic led to a multitude of changes in mental health care including the expansion of virtual care and telehealth services, which may have altered access to different provider types.23 For example, virtual care may have reduced some geographic barriers while potentially introducing new challenges related to technology access, digital literacy, and privacy. Despite these changes, many structural features of the Canadian mental health system such as differences in public versus private funding and provider availability remain largely unchanged. Because of this the patterns observed in this study likely reflect underlying systemic dynamics that continue to shape access to care. Likewise, these findings provide an important baseline for understanding how mental health service use has evolved in the post-COVID-19 pandemic era.”

Reviewer Comment: The assent/consent process for youth aged 12-18 needs to be explained.

Author Response: Data collection for the CCHS is conducted by Statistics Canada, which obtains informed consent from all participants. We have added: “For respondents aged 12–17 years, participation requires parental or guardian consent in addition to youth assent, in accordance with Statistics Canada data collection procedures.”

Reviewer Comment: Were response options for the outcome variable limited to the 4 provider types? Were there others? Or was it a free response that then was coded by researchers?

Author Response: The outcome variable was based on predefined response categories in the Canadian Community Health Survey (CCHS), where respondents were asked whether they had seen or talked to specific types of health professionals about their mental health in the past 12 months. The CCHS includes multiple provider categories (i.e., family doctors, psychiatrists, psychologists, social workers, nurses, and other health professionals).

This section now reads:“Response options in the CCHS are provided as predefined categories for different provider types (i.e., family doctor, psychiatrist, psychologist, social worker, nurse, and other health professionals), and respondents may select multiple providers. This analysis focused on the four most common providers for mental health care (family doctor, psychiatrist, psychologist, and social work) to ensure interpretability and comparability across cycles. Each provider was coded as a separate variable as respondents could see more than one provider.”

Reviewer Comment: How was “Visible minority” defined?

Author Response: In the CCHS, this variable is defined by Statistics Canada in accordance with the Employment Equity Act. The Employment Equity Act defines visible minorities as "persons, other than Aboriginal peoples, who are non-Caucasian in race or non-white in colour". The visible minority population consists mainly of the following groups: South Asian, Chinese, Black, Filipino, Arab, Latin American, Southeast Asian, West Asian, Korean and Japanese. We have added this clarification to the Methods section: “Visible minority status is defined by Statistics Canada, in accordance with the Employment Equity Act, as persons, other than Indigenous peoples, who are non-Caucasian in race or non-white in colour, based on self-identification.”

Reviewer Comment: I did not see any sample descriptives. This basic information is needed for readers to contextualize the findings.

Author Response: We have added a table with sample descriptives and the accompanying text: “Table 1 presents the weighted characteristics of respondents who reported consulting a mental health professional in the past 12 months. The sample was predominantly women (65.9%) and individuals aged 18–49 years. Most respondents reported higher household incomes (≥$80,000; 60.1%) and post-secondary education (87.1%). The majority identified as non-visible minorities (86.2%) and non-immigrants (83.7%), with 5.0% identifying as Indigenous. Most participants reported good or very good general (67.5%) and mental health (62.2%).”

Reviewer Comment: The authors interpret the pattern of service use over the years as “relatively stable.” is this a subjective observation or were differences between years tested? The latter would strengthen the manuscript.

Author Response: Thank you for this comment. We agree that the phrasing could imply formal statistical testing. We have revised the manuscript to clarify that interpretations of stability are based on descriptive trends rather than formal tests of differences across years. This clarification has been added to both the Results and Limitations sections.

The results now read: “Across all cycles, the rank order of providers was stable and the patterns of service use appeared relatively stable over time based on descriptive trends, although no formal statistical tests of differences across years were conducted.”

In the limitations we added: “Furthermore, we did not formally test differences in service use across survey years a result interpretations of temporal trends are based on descriptive patterns.”

Reviewer Comment: Table 1 is extremely hard to read and is not formatted following APA guidelines

Author Response: Thank you for this comment. We have reformatted Table 2 to improve readability and align with APA guidelines. Specifically, we combined adjusted odds ratios and confidence intervals into a single column, removed p-value columns, and simplified the table structure.

Reviewer Comment: Several assertions are made in the Discussion with no accompanying reference.

Author Response: Thank you for the comment we have added reference throughout where appropriate.

---

## [Editor Report · Decision Letter 2]

30 Apr 2026

Sociodemographic and Mental Health Predictors of Mental Health Service Use Across Provider Types

PONE-D-25-27752R2

Dear Dr. Pang,

We’re pleased to inform you that your manuscript has been judged scientifically suitable for publication and will be formally accepted for publication once it meets all outstanding technical requirements.

Kind regards,

Gerard Hutchinson, MD

Academic Editor

PLOS One
---

## [Editor Report · Acceptance letter]

PONE-D-25-27752R2

PLOS One

Dear Dr. Pang,

I'm pleased to inform you that your manuscript has been deemed suitable for publication in PLOS One. Congratulations! Your manuscript is now being handed over to our production team.

Kind regards,

on behalf of

Dr. Gerard Hutchinson

Academic Editor

PLOS One